# Statistical and machine learning models for predicting university dropout and scholarship impact

Stephan Romero⦾, Xiyue Liao ⓘ⦾*

Department of Mathematics and Statistics, San Diego State University, San Diego, California, United States of America

⦾ These authors contributed equally to this work.

\* xliao@sdsu.edu

**Data availability statement:** The data set is available from the public repository: UC Irvine Machine Learning Repository. The URL is https://archive.ics.uci.edu/dataset/697/predict+students+dropout+and+academic+success.

## Abstract

Although student dropout is an inevitable aspect of university enrollment, when analyzed, universities can gather information which enables them to take preventative actions that mitigate dropout risk. We study a data set consisting of 4,424 records from a Portuguese higher institution. In this study, dropout is defined from a micro-perspective, where field and institution changes are considered as dropouts independently of the timing these occur. The purpose of this analysis is twofold. First, we aim to build predictive models to learn of the significant socioeconomic and academic features associated with students' dropout risk. Another goal is to understand the relationship between financial status and dropout, especially the causal effect of being a scholarship holder. Propensity score matching is conducted first with the training set to better estimate the causal effect of being a scholarship holder on dropout status while controlling confounding variables. The predictive classifiers evaluated are Lasso regression, generalized additive model, random forest, XGBoost and single-layer neural network. The XGBoost model has the highest F1-score 0.904. According to this model, the most important features predicting dropout status are the student's second semester grades and the number of units they are credited. Whether a student's tuition fee is up to date, whether they owe money to a debtor, whether they are scholarship holders, and students' age at enrollment are also found to be important features. The Generalized Additive Model (GAM) performs competitively and offers clear interpretability, revealing how changes in actionable variables influence dropout risk. It shows that receiving a scholarship leads to the reduction in the odds of dropping out by nearly 40%, or by 22.2% in terms of probability when holding other factors fixed. As the study is based on data from a single institution and time period, and unobserved confounders cannot be fully ruled out, results should be interpreted with caution.

**Funding:** The author(s) received no specific funding for this work.

## 1 Introduction

Understanding important factors that are associated with student dropouts from higher education is important to institutions and students. Colleges and universities have the responsibility to help students succeed. High dropout rates indicate that many students are not receiving the support they need, whether academically or financially. Detecting the factors can allow institutions to act early to control or reduce dropout rates, help students get a degree, and pave the way for a better career outlook. According to the Education Data Initiative [1], college dropouts are 20% more likely to be unemployed compared to those who complete their degrees. In addition to academic and financial challenges, students may also face structural barriers that contribute to dropout risk. For example, incomplete information about university degrees at the time of high school graduation can affect students' decision-making and lead to suboptimal academic decisions. Such mismatches may set the stage for disengagement and eventual dropout. From a broader perspective, student attrition also represents an inefficiency within the higher education system, reducing the effectiveness of institutional investment and planning [2].

Recent research in predicting college dropouts has been focusing on utilizing machine learning techniques for better prediction accuracy and finding factors affecting dropout risks. Some studies utilized machine learning models to analyze data from a specific university or multiple universities, while others took a meta-analytical approach, examining existing literature to identify the most commonly discussed factors. Niyogisubizo et al. [3] studied a data set collected from 2016 to 2020 at Constantine the Philosopher University in Nitra. They explored a two-layer ensemble machine learning approach which fed the output of binary classifiers: random forest and XGBoost into a feed-forward neural network to improve the prediction results. A high F1-score (0.95) was achieved by this approach. Ortiz-Lozano et al. [4] focused on investigating the optimal time and types of data needed to identify students who are at risk of dropping out. The study analyzed demographic and academic information from 935 first-year students at an engineering school in Spain with classification tree, which had good interpretability due to its decision tree graph. The prediction accuracy was 76%. The key finding was that academic performance data served as a strong predictor of student withdrawal. Their research supported conclusions in previous studies about supporting the need for an early first-year intervention to improve retention. Ortiz-Lozano et al. [5] analyzed academic data from 3,583 first-year students on the Business Administration (BA) degree at the University of Barcelona and identified that the percentage of failed or unattended subjects in the first semester plays a significant role in predicting dropout. Several machine learning algorithms were applied, namely, neural networks, random forest, and logistic regression. All models exhibited good prediction accuracy and sensitivity (correctly predicted dropout). The average accuracy and sensitivity on the test set were 77% and 69%. Results are validated by a larger data set from several degree programs at another university. This study only used academic data to develop the model. Grades, course participation, and the rate of failed or unattended courses during the first semester were the most significant predictor of dropout risk. Huynh-Cam et al. [6] sought to equip universities with valuable insights to implement early interventions aimed at reducing dropout rates. Their study utilized enrollment records from 2,412 first-year students at a private university in Taiwan. The researchers tested various machine learning models, including decision trees, logistic regression, and neural networks. Among these, the decision tree model outperformed other models, achieving an accuracy of 97.59%. Key predictive factors identified included student loans, father's occupations, mother's educational level, department, type of admission, school fee waivers, and primary sources of living. Kocsis et al. [7] reviewed and evaluated 95 studies published after 2012. They

did a thorough comparison of data mining algorithms that have been applied to educational data sets, and they found that decision tree, logistic regression and neural networks are most commonly used educational data mining algorithms. Grade point average, obtained credits and gender were found to be most decisive predictors of academic performance. Silva, et al. [8] carried out a systematic review of 52 articles and examined various demographic, academic and learning factors that influence students' retention and dropout in the higher education system. In their study, they pointed out the multi-causal nature of university dropout. The most predictive factors were students' average scores, gender, course grades, course, age, ethnicity and scholarship. With these identified factors, they planned a machine learning project for the University of Brasilia to help improve retention rates. Aina et al. [9] did a comprehensive review of socio-economic literature for the past three decades on the student determinants of tertiary education dropout and provided insights about the main trends in university dropouts. By synthesizing empirical findings within a theoretical framework, the authors examined how informative issues (emphasized by the economic literature) and relational elements (emphasized by the sociological literature) contribute to students' decisions to discontinue their tertiary education. The factors include student demographic characteristics, abilities and behavior, parental background and family networks, academic/social integration and institutional/goal commitment, tuition fees and financial/in-kind aid, etc. Logistic regression is the most commonly used model in the studies reviewed in this paper. Empirical evidence showed that older freshmen, minority students, and students from poorer families have a higher dropout probability. On the other hand, female students, students with closer ties with peers, and students in institutions with greater commitment are less likely to drop out. Higher tuition fees only affect low-income students and financial aid has a positive impact on student retention. Lorenzo-Quiles et al. [10] is another recent review study, which used different databases to find out the most appropriate authors on the topic of university dropout since 2018 and identify variables that can cause university dropouts. Identified causes are mainly psychological and social factors. Support from parents, teachers and institutions is often one determinant why students regain their motivation and stay in institutions rather than drop out. Self-esteem and self-concept that lead to school failure or frustration are also directly related to dropout. Venkatesan et al. [11] reviewed 36 studies publish between 2000 to 2023, which applied statistical and machine learning approaches to predict student dropout in education. It was found that Random Forest and Decision Tree are among the most widely used techniques in those recent studies. Prediction accuracy and Area Under the Curve (AUC) are the most frequently employed metrics to assess the performance of predictive models. This review also pointed out some unsolved problems such as data imbalance, lack of interpretability of machine learning models, and geographic disparities that might lead to new research in the future.

In this case study, we discuss a data set from a public university in Portugal with students' academic and socioeconomic information. This data set spanned the academic years between 2008/2009 and 2018/2019. It was collected and first discussed in Martins et al. [12]. Random forest and three boosting models were built to help tutors identify students at risk of dropout in this data set. However, in that study, every feature was treated as numeric, though some features, such as `scholarship holder`, are categorical. Besides, no feature screening was done before modeling, though some pairs had a very high correlation or redundant information. Finally, with all features being used for modeling, while feature importance was ranked, the study did not explore how key features specifically influence dropout risk. The research gap we aim to address centers on three aspects: 1) estimating causal effects of scholarship status on dropout using propensity score matching (PSM), 2) improving model interpretability in dropout prediction, and 3) evaluating the performance and utility of boosting methods,

which remain under-explored in recent literature on student dropout. While prior studies have identified factors associated with dropout risk, few have investigated the causal relationship between financial support and dropout. By applying PSM, we seek to isolate the effect of holding a scholarship, thereby moving beyond correlation-based findings. In addition, we explore the potential of machine learning models—particularly the Least Absolute Shrinkage and Selection Operator (Lasso) [13], Generalized Additive Models (GAMs) [14] and XGBoost [15] — to offer both interpretability and predictive strength. Logistic regression was commonly used in previous studies due to its ease of interpretation. However, when applied to datasets with a large number of predictors—as is often the case in dropout studies—logistic regression becomes susceptible to multicollinearity and overfitting, which can compromise both model performance and interpretability. The Lasso method is a shrinkage method to reduce the multicollinearity and overfitting problem in logistic regression. We explore the Lasso method to evaluate whether it can achieve both high predictive accuracy and maintain a transparent linear structure. GAM is a more flexible model to handle non-linear effects than Lasso. As a semiparametric method, GAM strikes a balance of flexibility and interpretability. We further demonstrate that boosting methods like XGBoost can yield high predictive performance and valuable insights into variable importance like Random Forest, offering a complementary perspective to traditional statistical approaches. Our study, based on data from 4,424 students enrolled in 17 undergraduate programs at a public university in Portugal, thus contributes substantively to the understanding of dropout dynamics.

The rest of this manuscript is organized as follows: we first present the background information of the data set, check data structure and pre-process features in Sect 2. In Sect 3, we explain models and define model evaluation metrics. We present the results of our analysis in Sect 4, and close with a discussion of main findings, limitations and future work in Sect 6.

## 2 Materials and methods

### 2.1 Data background

The data set used in this study is from a public university in Portugal: Polytechnic Institute of Portalegre. It was first presented and discussed in [12] and it is available at the public repository: 10.24432/C5MC89. The data set consists of 4,424 observations and 36 variables, including students' socioeconomic and academic information between the academic years of 2008/2009 and 2018/2019. It was created by merging several disjoint databases. The compiled data set includes students enrolled in 17 undergraduate degrees, such as agronomy, design, education, nursing, journalism, management, social service and technologies. Each record in the compiled data set represents an individual student and there is no missing value. In this study, dropout is defined from a micro-perspective, where field and institution changes are considered as dropouts independently of the timing these occur. In contrast, a macro-perspective might define dropout more leniently. For example, only students who leave the entire education system and won't return are considered as dropouts, and students who switch majors or transfer schools are still considered enrolled, etc.

### 2.2 Data preprocessing

**2.2.1 Re-grouping and ordering qualification features.** In the data set, the features `mother's qualification`, `father's qualification`, and `previous qualification` have many levels, which were labeled from 1 to 44. Quite a few levels are missing. For example, `mother's qualification`, there are no levels 7 and 8 but there are levels 6 and 9. The ordering of the levels seems to be unclear. To reduce the complexity of

these features and impose a more understandable ordering on the levels, we re-group levels of `mother's qualification`, `father's qualification`, and `previous quali-fication`. Specifically, levels of these features, which are unclear, are dropped from the data set. The kept levels are re-grouped by their associated degrees and typical years of schooling, i.e., "high school diploma, no college", "some college, no degree", "associate degree", "bachelor's degree", "master's degree", "doctoral degree", and "professional degree." Next, because `mother's qualification` and `father's qualification` are highly correlated, to reduce multicollearity, we create and use a new feature `parents' qualification` instead, which records the highest level of education of a student's `mother's qualification` and `father's qualification`.

**2.2.2 Standardizing and one-hot encoding features.** The continuous features $x$'s are scaled using min-max scaling:

$$x' = \frac{x - \min(x)}{\max(x) - \min(x)}. \tag{1}$$

The min-max scaling is a commonly used scaling method. It brings the range of each predictor to be between 0 and 1. When applying a regularization model like the Lasso model in Sect 3.1, standardization is important such that features' coefficients will not be shrunk differently simply because they have different units or ranges. The categorical features are one-hot encoded, i.e., each category in a categorical feature is coded as 0 or 1, and 1 is used whenever an observation is in that category.

**2.2.3 Removing the enrolled category in the response.** Since we aim to predict whether a student dropped out or successfully graduated from college, we decide to remove observations with the target class "enrolled" from the data set. The final data set only include observations with the target class "graduated" or "dropout."

**2.2.4 Feature screening.** Curricular units from the first and second semesters are highly correlated as shown in Table 1. We keep the units' variables in the 2nd semester and remove the units' variables in the 1st semester, namely, the `curricular units 1st sem (credited)`, `curricular units 1st sem (enrolled)`, `curricular units 1st sem (approved)`, `curricular units 1st sem (grade)`, and `curricular units 1st sem (evaluations)` are removed. The features `nationality` and `international` have very high correlation and overlapping information, and we remove `nationality`.

A chi-squared test of independence is performed between the outcome of interest: dropout status and each categorical feature. It is followed by a post-hoc Bonferroni adjustment to determine if the dropout status is marginally related to each categorical feature. If the result of the test is statistically significant, i.e., $p$-value $<0.05$, then we keep this feature for modeling. For continuous features, the filtering method is the pairwise $t$-test with Bonferroni adjustment. The features `educational special needs`, `unemployment rate`, `GDP`, and `parent's qualification` are dropped from the data set. After the preprocessing

**Table 1. Pairs of features with Pearson's correlation coefficients >0.8.**

| | | |
|---|---|---|
| curricular units 1st sem (credited) | curricular units 2nd sem (credited) | 0.96 |
| curricular units 1st sem (enrolled) | curricular units 2nd sem (enrolled) | 0.94 |
| curricular units 1st sem (approved) | curricular units 2nd sem (approved) | 0.92 |
| curricular units 1st sem (grade) | curricular units 2nd sem (grade) | 0.85 |
| curricular units 1st sem (evaluations) | curricular units 2nd sem (evaluations) | 0.80 |
| nationality | international | 0.91 |

step, we finally end up with a data set with 3,400 observations and 25 variables for propensity score matching or modeling. S1 Table and S2 Table summarize the continuous and categorical features kept for this study. Portuguese universities grade individual courses (also called curricular units) using a numeric scale from 0 to 20. The admission and previous qualification grades are on a 0 to 200 scale, consistent with the Portuguese higher education admission system. Several variables describe a student's academic activity in a given semester: the number of enrolled units reflects courses registered for; evaluated units indicate those in which at least one assessment was taken; approved units represent courses passed. Credited units may stem from prior recognition or transfer. The grade variable summarizes average performance across evaluated courses. These indicators provide complementary perspectives on academic engagement and success.

**2.2.5 Propensity score matching.** In this study, a randomized controlled trial (RCT) is not feasible or inappropriate. For example, if a scholarship is aimed to be provided to students from low-income families or minority groups, then randomly issuing or denying the scholarship to qualified students would raise ethical concerns. Therefore, we find it necessary to employ a quasi-experimental approach to avoid ethical issues of random assignment, such as propensity score matching, to mimic the effects of RCT and to explore the influence of scholarship on dropout risks. Developed by Rosenbaum and Robin [16], propensity score matching is a method used to estimate the causal effect of a binary treatment variable or main predictor of interest, by achieving covariate balance in treatment groups. In an observational data set, it is common that covariates relevant to both the outcome variable and the treatment variable are unbalanced or have different distributions among treatment groups. Because of such confounding effects, it is hard to have a valid causal treatment effect estimate. When distributions of covariates are balanced within treatment groups, the casual effect of the main predictor can be better estimated.

The R package `MatchIt` [17] is used to perform propensity score matching. The feature `scholarship holder` serves as our main predictor or the treatment variable, where "dropout" represents the treatment group and "graduated" represents the control group. The following variables covariates were included in the propensity score estimation 1) demographic factors: age at enrollment, gender, marital status, displaced status (whether the student relocated from their residence to attend the university) 2) academic background: previous qualification, previous qualification grade, admission grade, application order 3) academic performance and engagement: curricular units enrolled, credited, evaluated, and graded across semesters; curricular units without evaluations 4) socioeconomic indicators: tuition fee payment status, debtor status, mother's and father's occupations 5) administrative and institutional factors: course of study, daytime/evening attendance, application mode. 6) national-level variable: with the exception of the inflation rate, all variables used in our analysis are measured at the individual level. The inflation rate is used solely as a macro-economic covariate in the propensity score matching procedure and is not included in the models predicting dropout. These covariates are selected based on their practical relevance and prior empirical evidence showing associations with both scholarship allocation and dropout risk [9,11]. Academic performance indicators, e.g., admission grade, prior qualifications, course grades, are typically used in scholarship decisions and are strong predictors of academic persistence. Socioeconomic variables such as tuition payment status, debtor status, and parental occupations reflect financial stability, which may influence both eligibility for financial aid and the ability to continue education. Demographic and institutional factors, including age, gender, course type, and attendance mode, are also known to correlate with both the likelihood of receiving support and the risk of dropout. Together, these factors

represent key dimensions (academic, economic, and demographic) commonly linked to both treatment and outcome in higher education research.

The `MatchIt` package offers several methods of calculating a propensity score, which is the probability of being in a treatment group or a control group, conditional on the covariates. If covariates for an individual in the treatment group and an individual in the control group are balanced, their propensity scores should be close, and we can make them a matched pair. We use logistic regression to estimate propensity scores. After a propensity score for each subject is computed, subjects from the treatment group and the control group are matched. Several matching methods are included in `MatchIt` determining the closeness of propensity score, including "nearest neighbor", "optimal", "full", and "subclass." To choose among the options, we assess covariate balance before and after matching using standardized mean differences (SMDs) and variance ratios. The matching method that results in the best balance is the "subclass" method. As shown in Fig 1, matching substantially reduced imbalance: after matching, nearly all covariates had absolute SMDs below 0.1, and variance ratios were within an acceptable range (most between 0.8 and 1.25). These diagnostics indicate that our

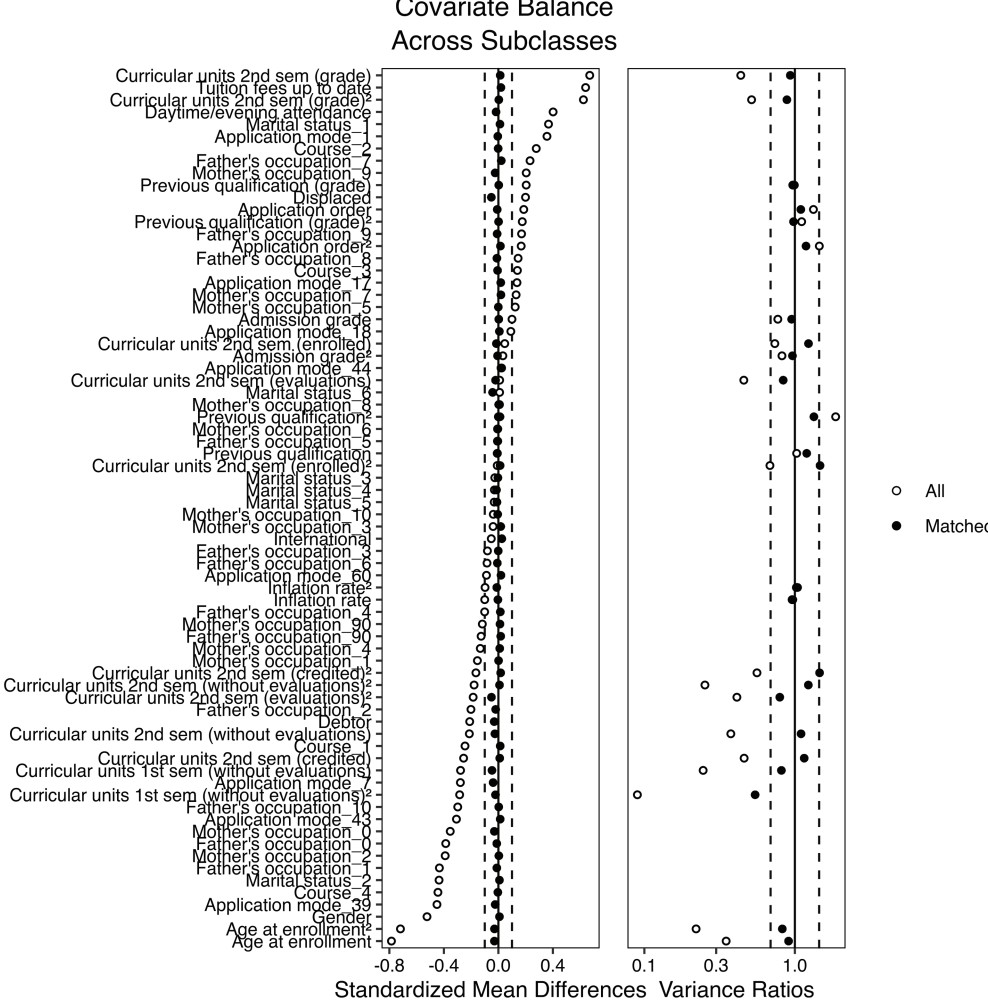

**Fig 1. The standardized mean differences and variance ratios of covariates before matching: ○ and after matching: ●.**

matching procedure achieved strong balance between treated and control groups, improving the credibility of the causal estimates. Details of theoretical support of these diagnostic tools can be found in [18]. However, we want to caution that when using propensity score matching, unobserved confounders, such as family expectations or mental health, may bias causal estimates if they influence both scholarship receipt and dropout risk. Regression Discontinuity Design (RDD) proposed in [19] offers a viable alternative, as it relies on a known cutoff for treatment assignment and assumes local similarity around the threshold. For example, if a scholarship is awarded based on income, RDD can estimate the causal effect by comparing students just below and above the income cutoff. This method avoids reliance on full covariate adjustment and may better approximate a randomized experiment. A relevant case study discussing the causal effect of financial aid on university attrition can be found in [20]. Future research could explore such designs in more detail using richer datasets.

## 3 Models

In this section, we briefly describe models used in this study. Lasso and GAM models are parametric and semi-parametric, which provides coefficient estimates for features and we use them to gain insights into the nature of the relationship between important features and the outcome. Random forest, XGBoost and artificial neural network are machine learning models, which usually have good prediction accuracy but lacks interpretability.

### 3.1 Lasso regression

Logistic regression is a generalized linear regression model using a logistic function to estimate the occurring probability of the event of "success", i.e., dropout happens. Typically, the event of success is coded as 1; the event of failure is coded as 0. The logit link function connecting the occurring probability of success: $P(Y_i = 1 | \mathbf{x}_i)$, with the predictors $\mathbf{x}_i = [x_{i1}, x_{i2}, \ldots, x_{ip}]^\top$, is defined as

$$\log\left[\frac{P(Y_i = 1 | \mathbf{x}_i)}{1 - P(Y_i = 1 | \mathbf{x}_i)}\right] = \beta_0 + \sum_{j=1}^{p} \beta_j x_{ij}, \ \ i = 1, 2, \ldots, n. \tag{2}$$

Deviance is a goodness-of-fit statistic often used for logistic regression [21]. In a binary classification task, Lasso adds an $L_1$ regularization or penalty term to the deviance of the logistic regression model in (2). The coefficient estimates $\hat{\beta}_\lambda^L$ minimize

$$\frac{1}{n}\text{Deviance}(\beta_0, \beta_1, \ldots, \beta_p) + \lambda \sum_{j=1}^{p} |\beta_j|. \tag{3}$$

$L_1$ regularization can force some coefficient estimates to be 0, and therefore it conducts variable selection and leads to a more interpretable model. The R package `glmnet` [22] is used for fitting and tuning the penalty term $\lambda$ of the Lasso model.

### 3.2 Generalized additive model (GAM)

Generalized additive models (GAM) [14] are an extension of the generalized linear model framework, where the systematic component $g[\mathbb{E}(Y)]$ can be expressed as:

$$g[\mathbb{E}(Y_i)] = \beta_0 + \sum_{j=1}^{p} s_j(x_{ji}) + \sum_{l=1}^{k} \beta_l z_{li}, \ \ i = 1, 2, \ldots, n. \tag{4}$$

Here, $s_i$'s of continuous feature $x_i$'s, which are smooth components fitted by penalized smoothing splines; $z_l$'s are categorical predictors, each of which has a coefficient estimate $\beta_l$. The function $g$ is known as the "link" function connecting the population mean $\mathbb{E}(Y)$ with the linear combinations of $s_i$'s. $g$ is used to define the distribution of $\mathbb{E}(Y)$. For example, when $g$ is the identical function, $\mathbb{E}(Y)$ is assumed to follow a Gaussian or normal distribution. Such a smoothing model allows for flexibility when non-linear relationships are more complicated than a parametric quadratic or cubic form between the systematic component, which is the left-hand side of Eq (4), and predictors $x_{1i}, x_{2i}, \dots, x_{pi}$.

The **R** package **mgcv** [23] is used for fitting a GAM model and choosing the penalty term in $s_j, j = 1, 2, \dots, p$, by generalized cross validation. One advantage of a GAM model when compared with machine learning methods is its statistical inference and interpretation because of its additive nature and the coefficient estimate for categorical features.

### 3.3 Random forest

Random forest is a decision-tree based ensemble machine learning method and can provide improvement of model generalization by de-correlating the trees. Tree-based methods involve segmenting the predictors into several simple regions, which are also called nodes. A top-down approach, successive splitting, is used to divide the predictors' space into $J$ non-overlapping regions. Decision trees often suffer from the over-fitting problem. Random forest improves decision trees by choosing a random subset of all candidate predictors when splitting a tree, which de-correlates the trees. Moreover, random forests are developed by fitting a number of decision trees and averaging the predicted values of each individual decision tree to make a prediction, which reduces variance of predictions [24]. The **R** package **randomForest** [25] is used to build this model.

### 3.4 Extreme gradient boost (XGBoost)

Boosting is another tree-based ensemble machine learning method. It improves the predictions of a decision tree by growing a tree sequentially, i.e., fitting a tree using residuals obtained from the previously trained trees. Each tree can be rather small with just a few terminal nodes [14]. XGBoost is a regularized form of gradient boosting and it reduces overfitting. The **R** packages **xgboost** developed by Chen and Guestrin [15] is used. It builds decision trees in a parallel pattern, instead of sequentially like regular boosting. This allows fast training if the user has access to a computation platform that allows parallel computing.

### 3.5 Single-layer neural network

Artificial neural networks (ANN) are computing systems that attempt to emulate neural networks in biological systems, specifically to the human brain [26]. ANNs are based on neurons, which are defined as atomic parts that compute the aggregation of their input to an output based on an activation function. An activation function defines the output of that node given a set of inputs. A single-layer ANN model for a binary classification task can be defined as:

$$\mathbf{z} = \beta_0 + \sum_{k=1}^{K} \beta_k A_k \tag{5}$$

and we use the sigmoid activation function to map $\mathbf{z}$ to the probability of a student's dropping out:

$$P(\mathbf{Y} = 1 | \mathbf{x}) = \frac{\exp(\mathbf{z})}{1 + \exp(\mathbf{z})}, \tag{6}$$

where $A_k = g\left[w_{k0} + \sum_{j=1}^{p} w_{kj}x_j\right]$, $K$ is the number of hidden units, $w_{kj}$'s are our weights that are adjusted through training, and $g(\cdot)$ is the activation function mapping features $\mathbf{x}_j$'s to each hidden unit $A_k$. Here we used the ReLU function, which is defined as $g(\cdot) = \max(0, \cdot)$. The R package `nnet` [27] is used to build the neural network model.

## 3.6 Parameter tuning

The data set is randomly split into a training set (80%) and a test set (20%). The models are built using the training set, in which covariates have been balanced by propensity score matching. Hyperparameters tuned by 5-fold cross validation during training are summarized in Table 2. The **caret** package [28], is used to tune these parameters.

## 3.7 Evaluation metrics

To evaluate the performance of each model, we use the test set to compute several metrics commonly used for binary classification tasks, which are defined and explained in Sect 3.7.1 to Sect 3.7.4. For each metric, we use the following terms: true positive (TP), false positive (FP), true negative (TN), and false negative (FN). A positive case represents a dropout case. Each metric ranges from 0 to 1, inclusively. A larger value implies better performance.

**3.7.1 Accuracy.** Accuracy is the proportion of correct predictions out of all predictions. It is defined as

$$\text{Accuracy} = \frac{\text{TP} + \text{TN}}{\text{TP} + \text{TN} + \text{FP} + \text{FN}}. \tag{7}$$

**3.7.2 Recall.** Recall measures how many out of all positive cases are predicted correctly. It is defined as

$$\text{Recall} = \frac{\text{TP}}{\text{TP} + \text{FN}}. \tag{8}$$

**3.7.3 Precision.** Precision is the ratio of correctly classified positive cases to all predicted positive values. It is defined as

$$\text{Precision} = \frac{\text{TP}}{\text{TP} + \text{FP}}. \tag{9}$$

**3.7.4 F1-score.** F1-score (10) is the harmonic mean of the precision (9) and recall (8) of a classification model. It is used if researchers care about precision and recall equally.

$$\text{F1-Score} = 2 * \frac{\text{Recall} * \text{Precision}}{\text{Recall} + \text{Precision}}. \tag{10}$$

**Table 2. Hyperparameter summary for each of the models.**

| Model | Description | Values |
|---|---|---|
| Lasso | $L_1$-penalty (`lambda`) | a grid spanning [0.01,1] |
| Random Forest | number of variables sampled at each split (`mtry`) | {2, 4, 6, 8, 16} |
| XGBoost | number of decision trees in the model (`nrounds`) | {10, 50, 100, 200} |
| | maximum depth of tree (`max_depth`) | {2, 4, 6} |
| | learning rate (`eta`) | {0.01, 0.1, 0.2} |
| | minimum loss reduction (`gamma`) | {0, 0.5, 1.0} |
| Neural Net | number of hidden units (`units`) | {8, 16, 32, 64, 128} |

**3.7.5  Area under ROC curve.**   A receiver operating characteristic (ROC) curve is created by plotting true positive rate against false positive rate. All classification thresholds from 0 to 1 are considered in such a plot. AUC is the area under the ROC curve, and it represents the degree of separability of a model. When a classification model makes random guesses, AUC will be 0.5. When AUC gets closer to 1.0, the model is better at predicting positive cases as positive and negative cases as negative.

# 4 Results

## 4.1 Model evaluation results

With the optimal tuning parameters based on the training set, each model is evaluated with the test set. Accuracy may not be the most useful metric when the outcome is imbalanced. In this data set, the ratio of graduate and dropout observations is about 1.5, which shows a slight imbalance. In terms of correctly separating two classes in the outcome, the random forest model has the highest AUC value 0.935. The F1-scores of random forest, XGBoost and GAM are all around 0.9 with XGBoost having the highest F1-score, showing that they detect the dropout cases effectively. ROC curves visualizing the AUC value for each model are included in Fig 2. Evaluation metrics are shown in Table 3.

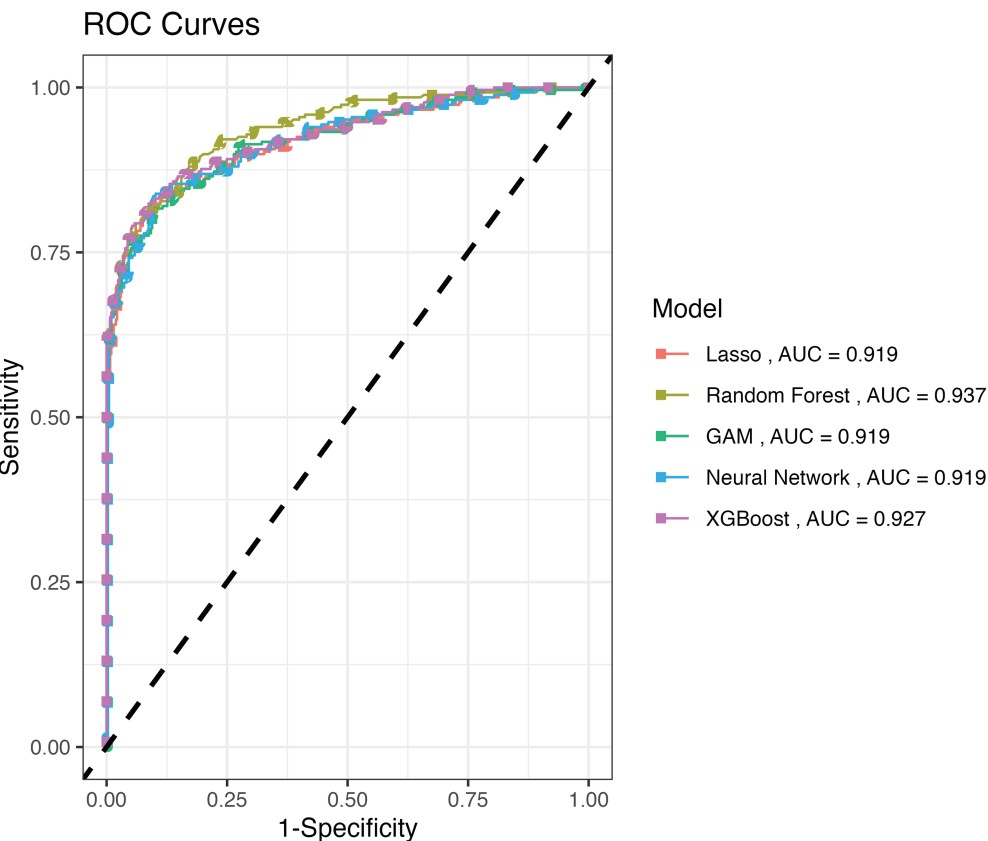

**Fig 2. ROC curves for the five models.**

**Table 3. Model evaluation results.**

| Model | Accuracy | AUC | F1-Score | Recall | Precision | Time (seconds) |
|---|---|---|---|---|---|---|
| Lasso | 0.875 | 0.919 | 0.834 | 0.798 | 0.873 | 0.91 |
| Random Forest | 0.879 | **0.935** | 0.904 | **0.872** | 0.939 | 4.14 |
| XGBoost | **0.882** | 0.924 | **0.907** | 0.871 | **0.947** | 348.60 |
| GAM | 0.868 | 0.919 | 0.895 | 0.863 | 0.930 | 134.00 |
| Neural Network | 0.862 | 0.912 | 0.891 | 0.854 | 0.932 | 484.20 |

## 4.2 Feature importance of the XGBoost model

Fig 3 ranks features according to each feature's importance value generated by the XGBoost model. The boosting model computes feature importance scores by summing the reduction in the model's binary entropy loss function caused by splits on each feature across all trees. Features with higher importance scores have a greater impact on the model's performance. Here we present the top seven features, which includes age, three financial factors, and three academic factors. Their importance scores are shown in Table 4.

The confusion matrix in Fig 4 illustrates the predictive performance of the XGBoost model on the test set. Rows of this matrix represent the cases classified into the dropout and graduate groups, while the columns indicate the actual group membership of each case.

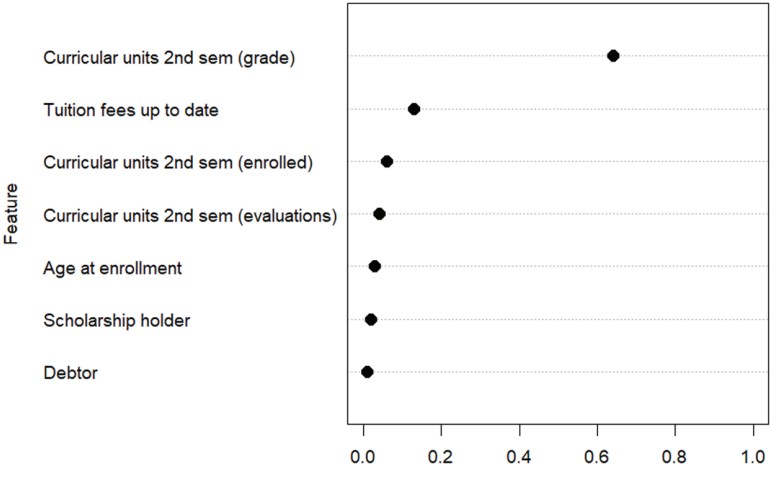

**Fig 3. XGBoost model feature importance ranking.**

**Table 4. XGBoost features with highest importance scores.**

| Feature | Importance score |
|---|---|
| Curricular units 2nd sem (grade) | 0.52 |
| Tuition fees up to date | 0.11 |
| Curricular units 2nd sem (enrolled) | 0.07 |
| Curricular units 2nd sem (evaluations) | 0.05 |
| Age at enrollment | 0.04 |
| Admission grade | 0.04 |
| Scholarship holder | 0.03 |

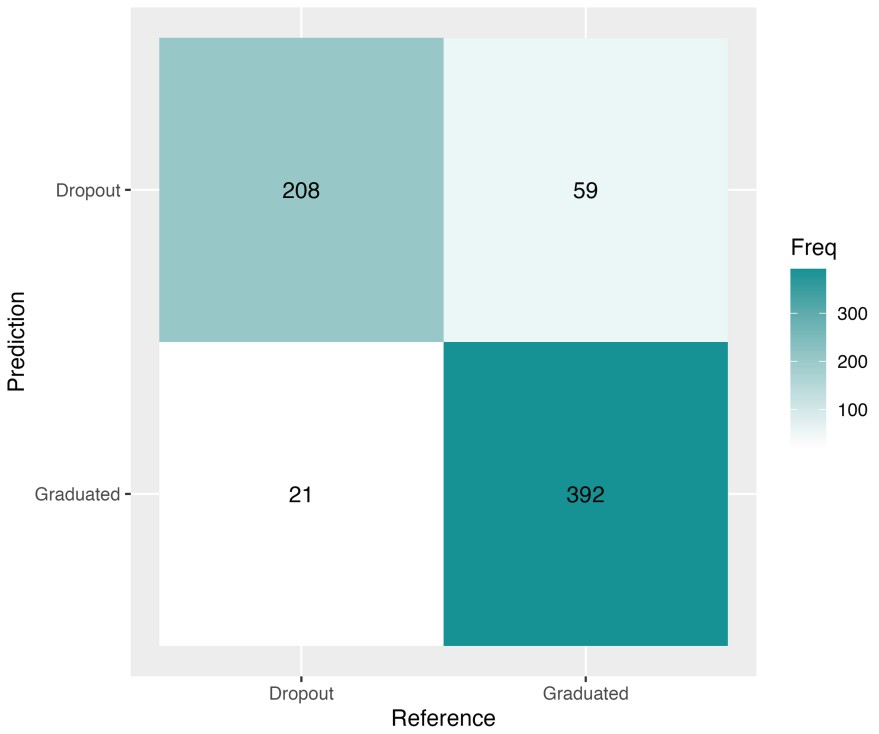

**Fig 4. XGBoost model confusion matrix for the test set.**

## 4.3 Significant features of the GAM model

### 4.3.1 Significant categorical features.
One advantage of the GAM model is that it is semiparametric and the nature of the relationship between important features and the outcome can be quantified by a coefficient estimate if the predictor is categorical; if the feature is continuous, then the estimated smooth component can be visualized and helps us understand how the feature affects the outcome. By the **R** package **mgcv**, we created Table 5, which shows significant categorical features ($p$-value <0.05) in the GAM model.

Table 5 shows that the effect of being a scholarship recipient is to decrease the odds of dropping out. To help interpret the model, we include both odds ratio and marginal effect,

**Table 5. Odds ratio (OR) and Marginal effect (ME) of significant categorical features of the GAM model.**

| Feature | Baseline group | OR | ME | $p$-value |
|---|---|---|---|---|
| marital status–married | single | 0.307 | -0.262 | 0.010 |
| marital status–divorced | single | 0.185 | -0.352 | 0.003 |
| application mode–2nd phase | 1st phase | 1.679 | 0.122 | 0.002 |
| application mode–3rd phase | 1st phase | 3.773 | 0.282 | <0.001 |
| course–health and human services | Business | 0.481 | -0.188 | 0.027 |
| mother's occupation–other | student | 0.089 | -0.447 | 0.025 |
| displaced–yes | no | 1.740 | 0.128 | <0.001 |
| debtor–yes | no | 5.918 | 0.353 | <0.001 |
| tuition fees up to date–yes | no | 0.086 | -0.424 | <0.001 |
| gender–male | female | 1.815 | 0.137 | <0.001 |
| scholarship holder–yes | no | 0.396 | -0.222 | <0.001 |

showing how dropout risk will change as levels of categorical predictors change while holding other predictors fixed. The odds of dropping out is about 40% lower, or equivalent, the probability of dropping out is 22% lower, if some student holds scholarship while other factors are fixed. The reasoning for this effect is twofold. First, a scholarship can be a source of motivation. Besides, many programs require recipients to maintain a minimum level of academic performance. Failure to meet these standards may result in the loss of ongoing scholarship eligibility. In some cases, when students' academic performance is extremely low or if they drop out, they may be required to repay funds. The student can feel responsible academically well to meet the scholarship bestower's explicit and perceived expectations. Second, a scholarship helps with any financial burden a student might be facing, thus mitigating the financial reasons for dropping out. We also observe that if a student owes money to a debtor, the odds of dropping out is six times higher. Students being displaced from their residence are about two times more likely to drop out. The reason for being displaced is unknown but it could be relevant to a student's financial situation.

**4.3.2 Significant continuous features.** Among the continuous features, we find that `admission grade`, `age at enrollment`, `curricular units 2nd semester (enrolled)`, `curricular units 2nd semester (evaluations)`, and `curricular units 2nd semester (grade)` are significantly related to dropout status.

In the GAM model in 4, we visualize the marginal effects of two most significant continuous predictors: age at enrollment and the number of graded curricular units in the second semester. These effects are computed as the estimated probability of dropout associated with each predictor, holding all other variables constant at their mean (for continuous variables) or median (for categorical variables). In Fig 5, it is shown that when age at enrollment increases, the probability of dropping out would increase and then decrease. The turning point happened around its median value: 20 years old. In Fig 6, it is shown that as graded curricular

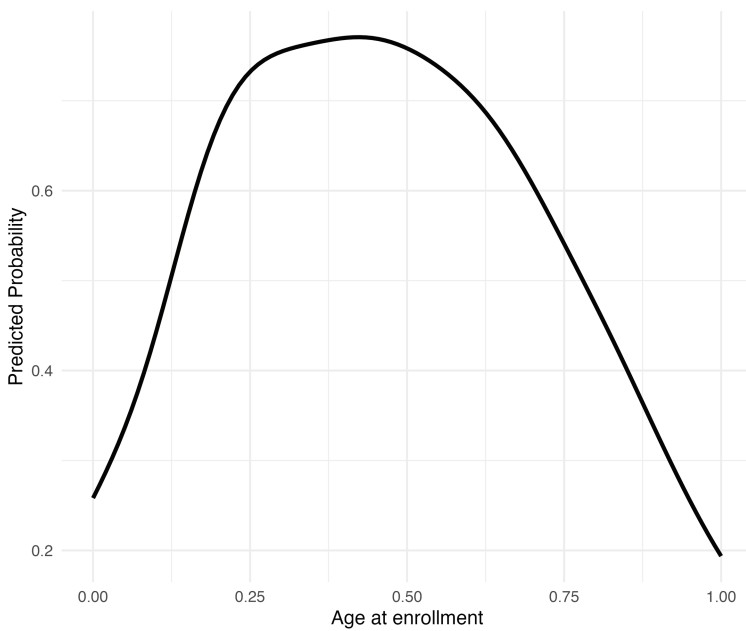

**Fig 5. Estimated marginal effect of age at enrollment.**

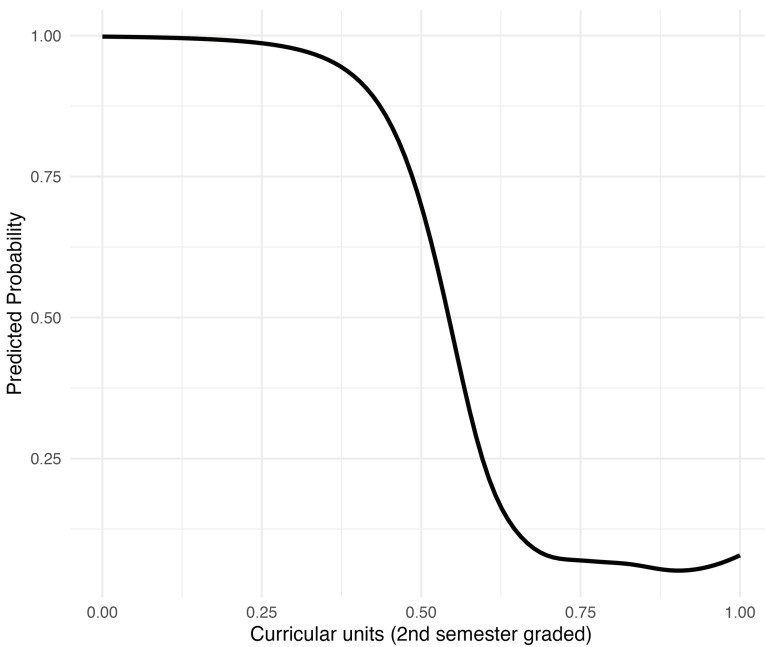

**Fig 6. Estimated marginal effect of curricular units 2nd semester (grade).**

units in the second semester increases, the chance of dropping out would steadily decrease. Note that we standardized each continuous feature, so the range of each feature in the plot is from 0 to 1. Similar plots showing the estimated marginal effects for admission grade, curricular units 2nd semester (enrolled), and curricular units 2nd semester (evaluations) are included in S1 Fig, S2 Fig, and S3 Fig in the Sect 6.

## 5 Discussion

This study contributes to the literature on student dropout by integrating machine learning models with quasi-experimental analysis to evaluate both associative patterns and potential causal effects, particularly concerning financial aid. The results consistently identified financial hardship, academic disengagement, and student background characteristics as key predictors of dropout. The importance of financial vulnerability was strongly supported by both the GAM model and XGBoost feature importance. Being in debt, having unpaid tuition fees, and lacking a scholarship emerged as the most predictive financial variables—aligning closely with recent findings that emphasize financial instability as a major driver of student dropout [9,11,20]. Our analysis using propensity score matching also showed that receiving a scholarship significantly reduces the likelihood of dropout, echoing prior research that highlights the retention benefits of financial support [10,20]. These results reinforce ongoing calls for institutions to prioritize financial aid accessibility, scholarship expansion, and debt management programs as central to student success strategies. Academic engagement was another critical dimension of dropout risk. Students failing to complete second-semester courses were substantially more likely to leave their studies – aligning with findings from [6], which emphasized that predictive models based on first-year performance data can help identify high-risk students early in their academic journey. From a policy perspective, the implications for universities are clear: tracking academic and financial indicators in real time

can allow institutions to identify students at risk and intervene before dropout occurs. For example, early alerts based on tuition payment status, course completion rates, or accumulated debt could initiate timely outreach through advising, tutoring, or emergency financial aid. This aligns with [10], who advocate for data-informed strategies that address both academic and socio-economic vulnerabilities.

This study has several limitations. First, the dataset used is five years old, and while it provides a robust historical overview, recent changes in higher education, such as remote learning or post-pandemic effects, may affect the generalizability of our findings. Second, the data were collected from a single public institution in Portugal. Institutional characteristics such as funding models, admissions policies, and student demographics differ widely across universities and countries. Thus, caution is warranted when generalizing these results to other contexts, especially outside of Portugal or within private institutions. Third, while propensity score matching adjusts for observed confounders, it cannot eliminate bias from unmeasured factors such as motivation, mental health, family pressure, or institutional support systems. These unobserved variables may influence both scholarship receipt and dropout, limiting the strength of causal inference. Finally, although the GAM model allowed for interpretability, it does not capture all possible interactions among important predictors.

In the future, improvements could be made by integrating data from multiple institutions or applying alternative causal inference techniques such as regression discontinuity design (RDD). RDD may be especially useful in settings where scholarship eligibility is based on a clear income or academic threshold, offering stronger quasi-experimental validity, as shown in [20]. Additionally, some potentially relevant variables, such as mental health, social integration, or employment status [10], were not available in our dataset. Integrating such information on student experiences could complement and enrich model-driven insights. When working with extremely large datasets, deep learning models, such as neural networks, may outperform GAM or XGBoost in predictive accuracy due to their ability to capture complex, nonlinear interactions among variables. Future researchers might also apply emerging techniques from explainable AI (XAI), such as SHapley Additive exPlanations [29], to help improve interpretability of deep learning models. This could allow institutions to benefit from both the accuracy of neural networks and the transparency needed for actionable insights.

## 6 Conclusion

This study evaluated five statistical learning models for predicting university dropout, using a dataset collected from a public university in Portugal. Among the models tested, XGBoost demonstrated the highest overall performance (Accuracy = 0.882, F1 = 0.907, Precision = 0.947). Key predictors of dropping out identified by XGBoost included the number of completed curricular units, up-to-date tuition payments, age at enrollment, and scholarship status. While Random Forest achieved the highest AUC (0.935), the AUC scores of the other models were comparably strong. The Generalized Additive Model (GAM) performs competitively in terms of standard evaluation metrics. More importantly, it offered clear interpretability, revealing how changes in actionable variables influence dropout risk. The findings confirm that both financial and academic factors are central to predicting student attrition. For example, students with unpaid tuition or existing debt face significantly higher dropout risk: being a debtor increases the odds of dropping out nearly sixfold (OR = 5.92), or equivalently, increases the probability of dropout by 35.3%, holding other factors constant. Conversely, students who keep their tuition fees up to date experience a 42.4% reduction in dropout probability. Using propensity score matching to adjust for covariate imbalance, the model also shows that receiving a scholarship leads to the reduction in the odds of dropping

out by nearly 40%, or by 22.2% in terms of probability. Beyond financial variables, academic and demographic factors also play a substantial role. Students enrolled in business programs face an 18.8% higher risk of dropping out compared to those in health or human services fields. Application timing is another important factor: students who apply during the second or third admission phases are more likely to leave their studies, exhibiting a 12.2% and 28.2% higher chance of dropping out. Being displaced was associated with a 12.8% increase in the likelihood of dropping out. Gender differences were observed as well, with male students exhibiting a 13.7% higher likelihood of dropping out. Marital status and parental education levels also influence dropout rates—single or never-married students are more at risk, and those whose mothers are currently students themselves show a stronger risk of student dropout. Among continuous variables, age at enrollment and the number of graded curricular units completed in the second semester emerged as the most influential predictors. Figs 5 and 6 illustrate their marginal effects. As shown in Fig 5, the probability of dropout initially increases with age at enrollment, peaking around the median age of 20, and then declines. Fig 6 highlights the heightened dropout risk among students demonstrating low academic engagement or performance.

Universities can act on these findings through targeted interventions. Early academic risk indicators, such as failing to complete second-semester courses, can trigger proactive advising or tutoring. Monitoring tuition status can help identify students in financial distress, who may benefit from emergency aid, payment plans, or financial counseling. Given the greater dropout risk among students without scholarships or those entering via later application phases, expanded scholarship access and tailored orientation programs could enhance student integration and persistence. Additionally, displaced students may require flexible academic policies and specialized support services. By combining accurate prediction with interpretable insights, this study offers a data-driven foundation for designing student support strategies that address both academic and financial vulnerabilities, which have been shown as two critical dimensions of university dropout.

## Supporting information

**S1 Table. Summary of continuous variables.**
(DOCX)

**S2 Table. Summary of categorical variables.**
(DOCX)

**S1 Fig. Estimated smooth component of admission grade.**
(TIFF)

**S2 Fig. Estimated smooth component of curricular units 2nd semester (enrolled).**
(TIFF)

**S3 Fig. Estimated smooth component of curricular units 2nd semester (evaluations).**
(TIFF)

## Author contributions

**Conceptualization:** Stephan Romero, Xiyue Liao.

**Data curation:** Stephan Romero.

**Formal analysis:** Stephan Romero.

**Investigation:** Stephan Romero.

**Methodology:** Xiyue Liao.

**Project administration:** Xiyue Liao.

**Software:** Stephan Romero, Xiyue Liao.

**Supervision:** Xiyue Liao.

**Validation:** Stephan Romero, Xiyue Liao.

**Visualization:** Stephan Romero.

**Writing – original draft:** Xiyue Liao.

**Writing – review & editing:** Xiyue Liao.

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
