## [Decision Letter · Decision Letter 0]

16 Oct 2024

PONE-D-24-42233Predicting university students' dropout using socioeconomic and academic featuresPLOS ONE

Dear Dr. Liao,

Thank you for submitting your manuscript to PLOS ONE. After careful consideration, we feel that it has merit but does not fully meet PLOS ONE’s publication criteria as it currently stands. Therefore, we invite you to submit a revised version of the manuscript that addresses the points raised during the review process.

We look forward to receiving your revised manuscript.

Kind regards,

Namal Rathnayake, Ph.D.

Academic Editor

PLOS ONE

Journal requirements: When submitting your revision, we need you to address these additional requirements. 1. Please ensure that your manuscript meets PLOS ONE's style requirements, including those for file naming. The PLOS ONE style templates can be found at https://journals.plos.org/plosone/s/file?id=wjVg/PLOSOne_formatting_sample_main_body.pdf and https://journals.plos.org/plosone/s/file?id=ba62/PLOSOne_formatting_sample_title_authors_affiliations.pdf 2. Please note that PLOS ONE has spec6ific guidelines on code sharing for submissions in which author-generated code underpins the findings in the manuscript. In these cases, all author-generated code must be made available without restrictions upon publication of the work. Please review our guidelines at https://journals.plos.org/plosone/s/materials-and-software-sharing#loc-sharing-code and ensure that your code is shared in a way that follows best practice and facilitates reproducibility and reuse. 3. We notice that your supplementary figures are uploaded with the file type 'Figure'. Please amend the file type to 'Supporting Information'. Please ensure that each Supporting Information file has a legend listed in the manuscript after the references list.

Additional Editor Comments:

The manuscript sounds good. However, please see the following comments.

1. There is a significant lack in the literature review. Please update a section for a literature.

2. At the end of the Lit, make sure to highlight the cons of the past studies and introduce your objectives.

3. You can add more illustrations to show the results, ex: Confusion matrix for predictions maybe?

Reviewers' comments:

Reviewer's Responses to Questions

**Comments to the Author**

1. Is the manuscript technically sound, and do the data support the conclusions?

Reviewer #1: Partly

Reviewer #2: No

2. Has the statistical analysis been performed appropriately and rigorously? 

Reviewer #1: Yes

Reviewer #2: Yes

3. Have the authors made all data underlying the findings in their manuscript fully available?

Reviewer #1: Yes

Reviewer #2: No

4. Is the manuscript presented in an intelligible fashion and written in standard English?

Reviewer #1: Yes

Reviewer #2: Yes

5. Review Comments to the Author

Reviewer #1: Review manuscript PONE-D-24-42233, entitled "Predicting university students' dropout using socioeconomic and academic features".

1. The study presents the results of original research.

The work presents original findings that advance the study of university dropout prediction. Its results align with the prevailing lines of academic inquiry, contributing further to the development of predictive models for university drop out.

2. Results reported have not been published elsewhere.

The study employs methodologies that have been applied and validated in other research. While the results have not been previously published, it should be noted that the authors need to improve and deepen their review of the literature. Some relevant studies, published in indexed journals on the subject, have been overlooked and could strengthen their research and results.

- Silva, E. C., Freitas, S., Soares Ramos, C., Muniz De Menezes, A. E., & Rodrigues De Araujo, L. K. S. (2023). A systematic review of the factors that impact the prediction of retention and dropout in higher education.

- Ortiz-Lozano, J. M., Aparicio-Chueca, P., Triadó-Ivern, X. M., & Arroyo-Barrigüete, J. L. (2024). Early dropout predictors in social sciences and management degree students. Studies in Higher Education, 49(8), 1303-1316.

- Ortiz-Lozano, J. M., Rua-Vieites, A., Bilbao-Calabuig, P., & Casadesús-Fa, M. (2020). University student retention: Best time and data to identify undergraduate students at risk of dropout. Innovations in education and teaching international.

- Huynh-Cam, T. T., Chen, L. S., & Lu, T. C. (2024). Early prediction models and crucial factor extraction for first-year undergraduate student dropouts. Journal of Applied Research in Higher Education.

3. Experiments, statistics, and other analyses are performed to a high technical standard and are described in sufficient detail.

Regarding this point, I believe the authors have done excellent work. The article presents a clear methodology, consistent with reference articles in the field. Statistical analyses are conducted at a high technical level and are described with sufficient detail.

However, I note that the database used is somewhat outdated, covering the period from 2008/2009 to 2018/2019. Given that more than five years have passed since the collection of these data, the conclusions may no longer fully reflect current conditions.

4. Conclusions are appropriately written and are supported by the data.

The conclusions are appropriately presented and supported by the data. The article successfully addresses its research objectives. However, the conclusions could be expanded, as not all results obtained are discussed.

The results section begins with a comparison of the evaluation of each statistical method used (Table 3), an important value-added aspect of the study. This helps readers better understand the statistical methodology employed.

In this case, the potential of Table 4 is not fully explored, as the discussion is based on only two or three variables, leaving others unaddressed, neither in the conclusions nor in the broader analysis (for example, the article barely touches on academic variables and their effect on school dropout, even though these variables are identified as the second most explanatory factor in the model).

The conclusions section should be expanded to include all results obtained from the study. I believe the authors have much more information to offer readers.

5. The article is presented in an intelligible fashion and is written in standard English.

The article presents some challenges in the presentation of tables and figures. There is a numbering overlap, with the same numbers assigned to tables within the text and to those in the appendix. Additionally, the figures in the appendix are not numbered. When the authors refer to these figures, they use terms such as "left" and "right," whereas assigning figure numbers would make referencing much clearer.

If there are figures that are not referenced in the text, they should not be included in the manuscript or its appendix.

6. The research meets all applicable standards for the ethics of experimentation and research integrity.

The research adheres to all applicable standards for ethical experimentation and research integrity.

Reviewer #2: The manuscript under review aims to address two objectives: (1) we aim to build predictive models to learn of the significant socioeconomic and academic features associated with students’ dropout risk, and (2) to understand the relationship between financial status and dropout, especially the causal effect of being a scholarship holder. Although these are important research purposes, there are several critical issues that preclude recommending the publication of this manuscript in its current form. Below, I present the main reasons for this decision.

Introduction

The manuscript describes the use of multiple predictive models (Lasso, Random Forest, XGBoost, GAM, and Neural Networks) to analyze dropout risk. However, there is no clear rationale for why these specific models were chosen over others, nor is there a thorough discussion of the existing literature on using these models for similar purposes. It is essential for the authors to situate their approach within the context of existing studies. For example, are there other studies that have used these models to predict dropout risk among university students? What are the strengths and limitations of these models as reported in the literature? Furthermore, there are known controversies and debates within the literature regarding the performance of these models in various contexts, which have not been adequately addressed. A more comprehensive literature review would help clarify why the selected models are appropriate for this analysis.

Please make the aim of the study clear in the introduction.

Lines 2-6: Please include references to support this explanation: “Understanding important factors in predicting college dropouts is not only important to 2 colleges and universities, but also to a nation. To colleges and universities, finding ways 3 to mitigate college dropout rates can lead to better utilization of funds for student 4 resources. To a nation, mitigating college dropout rates can potentially lead to a lower 5 unemployment rate and an increase in the nation’s workforce. “

Materials and methods

The manuscript fails to provide sufficient information about the study variables, which makes it difficult to understand, follow, and replicate the study. The authors mention that 36 variables were used in the analysis but do not explain what these variables are, how they were collected, or whether they include covariates.

Results

One of the aims of the study is to understand the causal effect of being a scholarship holder on dropout rates. However, the results presented in the manuscript do not provide clear insights into this relationship. There is no discussion on how the authors have addressed potential confounding factors, nor is there any robust analysis to determine causality.

The manuscript lacks a dedicated discussion section, which significantly weakens its contribution to the field. A discussion section is essential for interpreting the results, situating the findings within the broader context of the literature, and highlighting the practical implications of the study. Furthermore, it provides an opportunity for the authors to acknowledge the limitations of their work and suggest directions for future research. In the current manuscript, the results are presented without sufficient interpretation.

The limitations of the study, which should be included in the discussion, are part of the conclusion.

I recommend that the authors undertake a major revision before considering resubmission. Until these issues are adequately addressed, I cannot recommend this manuscript for publication.

6. PLOS authors have the option to publish the peer review history of their article (what does this mean?). If published, this will include your full peer review and any attached files.

Reviewer #1: No

Reviewer #2: No

---

## [Author Response · Author response to Decision Letter 1]

8 Dec 2024

We include a pdf file with nine pages to respond to reviewer and editor comments.

---

## [Decision Letter · Decision Letter 1]

24 Mar 2025

PONE-D-24-42233R1Predicting university students' dropout using socioeconomic and academic featuresPLOS ONE

Dear Dr. Liao,

Thank you for submitting your manuscript to PLOS ONE. After careful consideration, we feel that it has merit but does not fully meet PLOS ONE’s publication criteria as it currently stands. Therefore, we invite you to submit a revised version of the manuscript that addresses the points raised during the review process.

**Below, you will find the comments provided by external reviewers along with my own. By addressing these points, the paper will offer a more comprehensive and impactful analysis of university dropout factors, providing valuable insights for both researchers and practitioners in the field of education.**

We look forward to receiving your revised manuscript.

Kind regards,

Prof. Dr. Manuel Salas-Velasco, PhD

Academic Editor

PLOS ONE

**Additional Editor Comments:**

Abstract

The abstract provides a concise summary of the study's objectives, methodology, and key findings. However, it could be improved by clearly defining the term "dropout" and specifying the types of degrees considered (undergraduate, master's, or both). Additionally, the abstract should briefly mention the limitations of the study to provide a balanced overview.

Introduction

The introduction effectively highlights the importance of understanding dropout factors. However, it lacks a comprehensive literature review. The authors should include more recent studies and discuss the strengths and limitations of various predictive models used in similar research. This will help situate their work within the broader context of existing literature.

Methodology

- Definition of dropout. Clearly define what constitutes a dropout. Are students who switch majors or institutions considered dropouts? This distinction is crucial for the study's validity.

- Propensity Score Matching (PSM). Justify the choice of the "subclass" method over other methods like "nearest neighbor." Address recent criticisms of PSM and consider alternative methods like Coarsened Exact Matching (CEM).

- Confounding variables. Specify the confounding variables included in the PSM. Ensure these variables affect both the outcome (dropout) and the treatment (scholarship status).

Results

- Quantify the importance of features identified by XGBoost. Explain how each feature influences the dropout risk.

- Provide a more detailed interpretation of GAM results. Use marginal effects to explain the impact of significant predictors like scholarship status and debt on dropout probability.

- Include a confusion matrix for the XGBoost model to illustrate its predictive performance.

Discussion

The discussion section is currently missing. This section should interpret the results, compare them with findings from other studies, and discuss the practical implications for universities. Highlight the limitations of the study and suggest directions for future research.

Conclusion

The conclusion summarizes the key findings but should be expanded to include a discussion of all significant results. Address the implications of financial and academic factors on dropout rates and suggest specific interventions that universities can implement.

Limitations

The limitations section should be more comprehensive. Discuss the potential impact of using outdated data and the generalizability of the findings to other institutions and countries. Acknowledge the limitations of the predictive models used and suggest improvements for future research.

Additional Recommendations

- Data visualization. Use more visual aids like graphs and charts to present the results. This will make the findings more accessible and engaging.

- Practical applications. Provide concrete examples of how universities can use the study's findings to design interventions and support at-risk students.

Reviewers' comments:

Reviewer's Responses to Questions

**Comments to the Author**

1. If the authors have adequately addressed your comments raised in a previous round of review and you feel that this manuscript is now acceptable for publication, you may indicate that here to bypass the “Comments to the Author” section, enter your conflict of interest statement in the “Confidential to Editor” section, and submit your "Accept" recommendation.

Reviewer #1: All comments have been addressed

Reviewer #2: All comments have been addressed

Reviewer #3: (No Response)

2. Is the manuscript technically sound, and do the data support the conclusions?

Reviewer #1: Yes

Reviewer #2: (No Response)

Reviewer #3: Yes

3. Has the statistical analysis been performed appropriately and rigorously? 

Reviewer #1: Yes

Reviewer #2: (No Response)

Reviewer #3: Yes

4. Have the authors made all data underlying the findings in their manuscript fully available?

Reviewer #1: Yes

Reviewer #2: (No Response)

Reviewer #3: Yes

5. Is the manuscript presented in an intelligible fashion and written in standard English?

Reviewer #1: Yes

Reviewer #2: (No Response)

Reviewer #3: Yes

6. Review Comments to the Author

**Reviewer #1: **The authors have addressed all the reviewers' comments. They have revised the article, incorporating the main changes requested by the reviewers, which has led to a higher-quality work with greater scientific consistency and, ultimately, an improved final version.

**Reviewer #2: **(No Response)

**Reviewer #3:** First, the topic addressed in the manuscript is highly relevant. Given the current challenges faced by universities due to demographic shifts and declining enrollment rates, it is crucial to analyze the academic trajectories of students to better support them throughout their educational journey. The study’s focus on predicting university dropout is, therefore, well justified.

However, while the relevance of the topic is clear, certain aspects of the theoretical justification require further development. Specifically, the literature review lacks depth, and the manuscript does not clearly outline the main trends in university dropout. To strengthen the conceptual framework, the authors could consider incorporating key references such as Venkatesan et al. (2024), Aina et al. (2022), and Lorenzo-Quiles et al. (2023), which provide recent insights into this issue.

Additionally, the study’s contribution needs to be more explicitly defined. It would be beneficial for the authors to clarify the specific research gap they aim to address and to better articulate what the use of machine learning offers compared to traditional techniques such as logistic regression. Expanding on this point would reinforce the manuscript’s originality and its added value to the field.

Regarding the methodology, the authors should be cautious when discussing causality, as the study employs a quasi-experimental approach. The temporal sequencing and directionality of certain variables—such as scholarship allocation—are not clearly established. Since this is not a randomized controlled trial (RCT), the manuscript should acknowledge the limitations in making causal claims and consider alternative approaches to better approximate causal effects.

To strengthen the methodological rigor, the authors could discuss potential strategies to mitigate these limitations. Explicitly addressing these aspects would enhance the validity of the findings and provide a more nuanced interpretation of the results.

The discussion section requires greater depth and a more thorough interpretation of the results. While the main findings are presented, there is a lack of connection with existing literature, which limits the contextualization of the study’s contributions. The manuscript would benefit from a more detailed comparison with previous research on university dropout rates and the impact of socioeconomic and academic factors. Additionally, there is insufficient evidence provided to frame the findings within broader academic discussions, making it difficult to assess their significance beyond the analyzed dataset. Strengthening these aspects by incorporating relevant references and discussing how the results align or diverge from prior studies would enhance the clarity and impact of the discussion.

7. PLOS authors have the option to publish the peer review history of their article (what does this mean?). If published, this will include your full peer review and any attached files.

Reviewer #1: No

Reviewer #2: No

Reviewer #3: No

---

## [Author Response · Author response to Decision Letter 2]

3 Apr 2025

We includes responses to reviewer and editor comments in the file: responses_to_reviewers_R2.pdf.

---

## [Editor Report · Decision Letter 2]

2 Feb 2025

PONE-D-24-42233R2Predicting university students' dropout using socioeconomic and academic featuresPLOS ONE

Dear Dr. Liao,

Thank you for submitting your manuscript to PLOS ONE. After careful consideration, we feel that it has merit but does not fully meet PLOS ONE’s publication criteria as it currently stands.

The revised version improves substantially with the incorporation of the recommendations, but there are still some aspects that need to be addressed for further improvement of the paper.

Therefore, we invite you to submit a revised version of the manuscript that addresses the points raised during the review process.

We look forward to receiving your revised manuscript.

Kind regards,

Dr. Manuel Salas-Velasco, PhD

Academic Editor

PLOS ONE

**Additional Editor Comments:**

1. My first suggestion is to change the title. The current title does not reflect the complexity of the paper; it is very descriptive and more suited to an education-focused journal. The authors should consider alternative titles that incorporate both the models used and the main variable, which is being a scholarship holder. As a suggestion:

"Statistical and Machine Learning Models for Predicting University Dropout and Scholarship Impact"

2. Introduction (first paragraph). It is also important to highlight that incomplete information about university degrees at the time of high school completion is often an additional factor. This lack of clarity affects students' decision-making, frequently leading to suboptimal academic choices. Furthermore, within a higher education production model, student dropouts represent a significant inefficiency that undermines the overall efficiency of the higher education system. (Add reference / Fig. 1) https://doi.org/10.1007/s10671-019-09254-5

3. In section 3.2.3, the phrase “Because we are interested in predicting whether a student dropped out or graduated from their course” could indeed be rewritten for greater clarity for an international audience. A more precise alternative might be: “Since we aim to predict whether a student dropped out or successfully graduated from college.”

4. There are 25 final variables in the analysis, but the information presented in S1 (Table 4) is confusing for an international audience. Specifically, it should be clarified how many credits the courses in a semester have. The median value of 6 does not provide clarity. What is the grading system? A median score of 12.33 is not informative. Why is the inflation rate used? The data should be individual, corresponding to each student. When using group-level data, estimation issues arise, which can be addressed through multilevel regression. Overall, the table should explain what each variable measures.

5. In comment 5.3.1, "First, a scholarship can be a source of motivation," it should also be considered that students who do not meet the required academic performance lose their scholarship. In some cases, they may even be required to return the funds if their academic performance is extremely low or if they drop out.

---

## [Author Response · Author response to Decision Letter 3]

5 May 2025

We respond the editor's comments in a pdf file submitted as responses_R3.pdf.

---

## [Editor Report · Decision Letter 3]

6 May 2025

Statistical and Machine Learning Models for Predicting University Dropout and Scholarship Impact

PONE-D-24-42233R3

Dear Dr. Liao,

We’re pleased to inform you that your manuscript has been judged scientifically suitable for publication and will be formally accepted for publication once it meets all outstanding technical requirements.

Kind regards,

Dr. Manuel Salas-Velasco, PhD

Academic Editor

PLOS ONE

Additional Editor Comments:

In section 3.2.5, is the national-level variable supposed to be 6)?

There is a typo in Table 5—it should say “Business.”
---

## [Editor Report · Acceptance letter]

PONE-D-24-42233R3

PLOS ONE

Dear Dr. Liao,

I'm pleased to inform you that your manuscript has been deemed suitable for publication in PLOS ONE. Congratulations! Your manuscript is now being handed over to our production team.

Kind regards,

on behalf of

Dr. Manuel Salas-Velasco

Academic Editor

PLOS ONE